
# Seventeen years of ozone sounding at L'Aquila, Italy: evidence of mid-latitude stratospheric ozone recovery and tropospheric profile changes

Daniele Visioni[1*], Giovanni Pitari[2], Vincenzo Rizi[2,3], Marco Iarlori[2,3], Irene Cionni[4], Ilaria Quaglia[2], Hideharu Akiyoshi[5], Slimane Bekki[6], Neal Butchart[7], Martin Chipperfield[8], Makoto Deushi[9], Sandip S. Dhomse[8], Rolando R. Garcia[10], Patrick Jöckel[11], Douglas Kinnison[10], Jean-François Lamarque[10], Marion Marchand[6], Martine Michou[12], Olaf Morgenstern[13], Tatsuya Nagashima[5], Fiona M. O'Connor[7], Luke D. Oman[15], David A. Plummer[16], Eugene Rozanov[17], David Saint-Martin[12], Robyn Schofield[18], John Scinocca[16], Andrea Stenke[17], Kane Stone[18], Kengo Sudo[14], Taichu Y. Tanaka[9], Simone Tilmes[10], Holger Tost[11], Yousuke Yamashita[5], Guang Zeng[13]

[1] Cornell University, Sibley School of Mechanical and Aerospace Engineering, Ithaca, NY, USA

[2] Department of Physical and Chemical Sciences, Università dell'Aquila, L'Aquila, Italy

[3] CETEMPS, Università dell'Aquila, L'Aquila, Italy

[4] ENEA, Ente per le Nuove Tecnologie, l'Energia e l'Ambiente, Rome, Italy

[5] National Institute of Environmental Studies (NIES), Tsukuba, Japan
[6] LATMOS/IPSL, UVSQ Université Paris-Saclay, Sorbonne Université, CNRS, Guyancourt, France

[7] Met Office Hadley Centre (MOHC), Exeter, UK

[8] School of Earth and Environment, University of Leeds, Leeds, UK

[9] Meteorological Research Institute (MRI), Tsukuba, Japan

[10] National Center for Atmospheric Research (NCAR), Boulder, Colorado, USA

[11] Deutsches Zentrum für Luft- und Raumfahrt (DLR), Institut für Physik der Atmosphäre, Oberpfaffenhofen, Germany

[12] CNRM UMR 3589, Météo-France/CNRS, Toulouse, France
[13] National Institute of Water and Atmospheric Research (NIWA), Wellington, New Zealand

[14] Graduate School of Environmental Studies, Nagoya University, Nagoya, Japan

[15] NASA Goddard Space Flight Center (GSFC), Greenbelt, Maryland, USA

[16] Environment and Climate Change Canada, Montréal, Canada

[17] Institute for Atmospheric and Climate Science, ETH Zürich (ETHZ), Zürich, Switzerland

[18] School of Earth Sciences, University of Melbourne, Melbourne, Victoria, Australia

*Correspondence to*: Daniele Visioni (daniele.visioni@cornell.edu)

**Abstract.** Ozone profile measurements collected at L'Aquila (Italy, 42.4°N) during seventeen years of radio-sounding (2000-2016) are presented here, with an analysis of derived trends. Model results from the SPARC-CCMI exercise are used in parallel to highlight the physical and chemical mechanisms regulating mid-latitude ozone trends. The statistically significant trends highlighted in time series at L'Aquila are those in the mid-upper stratosphere (+5.9±4.2), mid troposphere



(+5.9±2.4) and upper troposphere (+2.5±0.9), all in percent/decade. The upper stratospheric positive trend was already well
documented in recent WMO assessments and attributed to the starting decline of stratospheric $Cl_y$ and $Br_y$ and to the
stratospheric cooling induced by increasing well mixed greenhouse gases, thus slowing down gas-phase reactions that
destroy ozone in the upper stratosphere. The ozone increase in the mid-upper troposphere is largely regulated by the
increasing strength of the Brewer-Dobson circulation, which moves more ozone from the tropics to the extratropics and
enhances the tropospheric influx from the lowermost stratosphere. This climate feedback mechanism on tropospheric ozone
is only partially compensated by the increasing chemical ozone loss associated to higher $H_2O$ values in response to the
tropospheric warming. We also note that ozone trends obtained in the lower stratosphere are negative (-2.2 percent/decade),
but do not result to be statistically significant in our analyses.

## 1 Introduction

Recent ozone assessments (WMO, 2011; WMO, 2014) have proved that the declining emission of ozone-depleting
substances (ODS) in compliance of the limitations imposed by the Montreal protocol has significantly decreased the
chemical destruction of stratospheric ozone. This has resulted in a strong reduction in the negative ozone trends observed
from 1970 to 1997 or even in the change of their sign, with emerging evidences of positive trends, at least in the upper
stratosphere (globally, Chipperfield et al., 2017, and with particular regard to the Antarctic, Strahan and Douglass, 2018).
Coordinated model inter-comparison initiatives, as SPARC-CCMVal (SPARC-WCRP, 2010) and SPARC-CCMI (Eyring et
al., 2013), made use of coupled chemistry-climate coupled models (CCMs) to better understand the complex
interconnections between ozone photochemistry and radiative-dynamical processes in the middle atmosphere, as well as
their evolution in time. These model inter-comparison projects have highlighted a robust scientific understanding of the
driving mechanisms of ozone depletion.
CCMs demonstrated a significant impact of ozone changes on stratospheric heating rates and circulation changes and also
showed how increasing greenhouse gases (GHGs) cool the stratosphere (Rind et al., 1990) and slow down homogeneous
chemical reactions that deplete ozone in the upper stratosphere (Bekki et al., 2013; Marsh et al., 2016). Finally, the model
simulations give a robust indication of a time-increasing strength of the Brewer-Dobson circulation (BDC) (Garcia and
Randel, 2008; McLandress et al., 2010; Chipperfield et al., 2013; Polvani et al., 2018), producing an ozone decrease in the
tropical lower stratosphere and an increase out of the tropics with respect to a control case with fixed GHGs.
The impact of an enhanced mid-latitude tropospheric influx from the lowermost stratosphere was discussed by Stevenson et
al. (2006). In that case, model simulations were designed with fixed surface emissions of tropospheric ozone precursors and
climate conditions projected for year 2030, in comparison with the standard case of year 2000 climate. They concluded that
two climate feedbacks, i.e., tropospheric water vapor increase and enhanced stratospheric input, appear to be the dominant



mechanisms operating on the budget of tropospheric ozone. The impact of stratospheric ozone on tropospheric ozone variability has been previously studied by Ordonez et al. (2007), Terao et al. (2009) and Hess and Zbinden (2011).

In this work we analyze the results of a time-series of ozone soundings taken in L'Aquila, Italy (42.3°N, 13.4° E) from the year 2000 to the year 2016 that have never been published before in the scientific literature. We first compare our results with available data from other ozonosondes in the same latitudinal band previously described and validated (Considine et al.,

2008; Tilmes et al., 2012). We then derive linear trends for different partial columns from our collected data spanning from the Boundary Layer to the mid-upper stratosphere and compare them with available simulations from the CCMI project, trying to determine the causes of the upper tropospheric and lower stratospheric recovery by analyzing the differences between the historical simulations (refC2) and sensitivity simulations where GHGs are kept fixed at 1960 levels (fGHG). Ozone sensitivity to greenhouse gases forcing in CCMI models has already been described in terms of global and

stratospheric changes for the 21st century (Morgenstern et al., 2018; Dhomse et al., 2018), while the inter-model spread in the tropospheric column in CCMs has been discussed by Yung et al. (2013, 2018) and Revell et al. (2018). Here, however, we focus on changes in the recent past (2000-2016) and in the mid-northern latitudes and try to understand the causes analyzing CCMI models results, focusing in particular on temperature and strat-trop fluxes changes, therefore looking at the differences with a simulation where no forcing from increased greenhouse gases is present.

## 80  1 Ozone sounding results and trend analysis

Since 1994, CETEMPS/DSFC at the of University of L'Aquila has operated an ozonesonde station (42.38N, 13.31E, site elevation 683m a.s.l.). In the period 2000-2016 the quality-checked balloon-borne ozone measurements conducted in troposphere and stratosphere were 295, with a pace of about 2 ozone-soundings per month; in some periods the observational activity was stopped or reduced for lack of funding (2002 and 2003), or because of a major earthquake (middle of 2009). The

ozone profile measurements are carried out with an ElectroChemical Cell (ECC, 6A Science Pump Corporation®) coupled to pressure, temperature, relative humidity sensors and GPS receiver (Vaisala® RS80, RS90, RS91, and RS41-SG) with the Vaisala® MW11, DIGICORA III and MW41 sounding systems along the different periods; the flying platform is a rubber balloon (Totex® 1200 g) able to reach altitudes above 10 hPa.

In order to homogenize the data (ozone profiles), the ECC sensors preparation's procedure accurately follows the

prescriptions about the sensor solution concentrations, the pump flow rate and the ECC background current measurements (Deshler et al., 2017). The vertical resolution of the raw ozone profile is better than 30 m, and the typical accuracies of pressure, temperature and relative humidity sensors are below 0.5 hPa, 0.3 K, and 2%, respectively. Concerning the ECC measurement uncertainties, following the procedure outlined in Witte et al. (2018), it is possible to state that the ozone partial pressure profile has an overall (uncorrelated) error between 4 and 12% in the troposphere, and about 4% in the

altitude range between 100 and 10 hPa levels. The various sources of systematic errors, that are also altitude dependent, increase the total error to about ±12% (Komhyr et al., 1995; Witte et al., 2018). Uncertainties of the total column of ozone



(TCO) are roughly less than 20 DU, that is about 5-8% of the TCO. The TCO is estimated using the ozone profile, plus the residual ozone (quantity from the maximum altitude of the flight (usually above 30 km or 10 hPa level) to the top of the stratosphere). This quantity is estimated assuming that the ozone has a constant mixing ratio equal to the measured value at
the top of the measured profile, up to a pressure of 0 hPa (as detailed in the Quality Assurance and Quality Control for Ozonesonde Measurements in GAW GAW Report- No. 201, WMO 2014b).

In the recent past the L'Aquila ozone-sounding data were used in the validation of stratospheric ozone profiles retrieved by the Michelson Interferometer for Passive Atmospheric Sounding (MIPAS), onboard ENVISAT (Rizi et al., 2002; Cortesi at al., 2007), and in the validation campaign of tropospheric ozone profiles estimated by the Ozone Monitoring Instrument
(OMI) data on the NASA-Aura platform (Di Noia et al., 2013).

### 2.1 Ozone profiles

The results of ozone soundings at L'Aquila from 2000 to 2016 are presented in Figs.1-4, in terms of the time-averaged vertical profiles, in log-pressure height coordinates, of mixing ratio (Fig. 1) and partial columns (Fig. 2-3), as well as time
series of the partial columns (Fig. 4). The vertical layers on which the partial columns are calculated are listed in Table 1; the basic statistics on these columns and on the calculated linear trends are presented in Table 2.

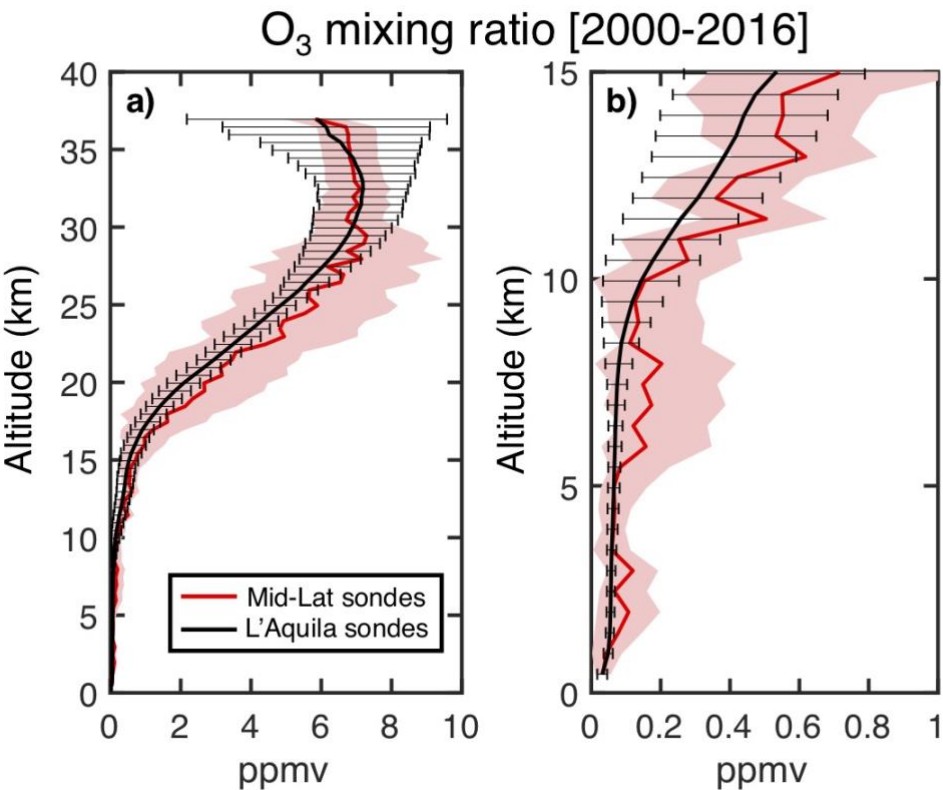

**Fig. 1.** Vertical profiles of the ozone volume mixing ratio observed at L'Aquila, Italy and averaged over the years 2000-2016 (black lines). Black whiskers represent ±1σ of the time variability. The red lines show the mixing ratio averaged over the same period of time from other northern mid-latitude ozone sounding stations, with the shading representing ±1σ of the variability of mean values among these 9 stations (Boulder, Hohenpeissenberg, Legionowo, Madrid, Payerne, Prague, Sapporo, Tsukuba, Wallop) (Tilmes et al., 2012). Panel (a) shows the complete profile extension over the troposphere and the stratosphere (0-10 ppmv), whereas panel (b) is a zooming over the troposphere (0-1 ppmv).

In Fig. 1, we show a comparison of the vertical profiles of the ozone volume mixing ratio in L'Aquila against an average over the same period of 9 other stations (Boulder, Hohenpeissenberg, Legionowo, Madrid, Payerne, Prague, Sapporo, Tsukuba, Wallop) that lie in the same latitudinal range (between 35° and 55° N). The ozonosondes data for the other soundings is available at the World Ozone and Ultraviolet Data Center (http://www.woudc.org/); a full climatology of these soundings for the period 1995-2011 is available from Tilmes et al. (2012) and Considine et al. (2008) for earlier periods. In Fig. 1a we show the entire profile from the ground to 37 km, while in Fig. 1b we only show the profile up to 15 km due to the different magnitude of the ozone values. In general, the ozone values measured in L'Aquila show a strong agreement



with the mean value derived from other sondes in the same latitudinal band, both at low altitudes with smaller values and in the stratosphere.

The different variabilities shown between the L'Aquila measurements (black bars) and the other measurements (red shading)
can be explained by considering that for the former, the error bars only represent the time variability while for the latter the shaded area represent the spread of the time-averaged results from the different soundings, i.e. some spatial variations in ozone. The difference can be seen in Fig. 2, where the same results are shown considering the partial ozone columns (in DU) with the altitude limits described in Table 1. In this case we show in Fig. 2a the variability defined as 1 standard deviation of the time average of the 9 stations, while in Fig. 2b the variability is defined as the standard deviation of the ensemble of
measurements of all 9 stations in that altitude range: this is much more comparable with the errorbars for the L'Aquila soundings calculated the same way. We further show in Fig. 3 the results for the single sondes location used in this paper. The larger errorbars in the uppermost stratosphere are a consequence of the decrease in data availability, since not all ozonosondes reach this altitude, and the values are not always as reliable as those at lower latitudes.

**Table 1.** Layers where partial $O_3$ columns are calculated. Z is the pressure altitude, defined as $H \times \log(p_0/p)$, where H is the atmospheric mean scale height (7 km), p is the pressure and $p_0$ the reference globally averaged surface pressure (1013 hPa).

| Partial column height (km) | Pressure (hPa) | Name of vertical layers |
|---|---|---|
| Z < 2.5 | p < 700 | Boundary Layer |
| 2.5 < Z < 7.5 | 700 < p < 340 | Mid Troposphere |
| 7.5 < Z < 12.5 | 340 < p < 170 | Upper Troposphere |
| 12.5 < Z < 17.5 | 170 < p < 80 | Uppermost Troposphere - Lowermost Stratosphere |
| 17.5 < Z < 22.5 | 80 < p < 40 | Lower Stratosphere |
| 22.5 < Z < 27.5 | 40 < p < 20 | Mid-Lower Stratosphere |
| 27.5 < Z < 32.5 | 20 < p < 10 | Mid Stratosphere |
| 32.5 < Z < 37.5 | 10 < p < 5 | Mid-Upper Stratosphere |

**Table 2.** Statistics of partial $O_3$ columns (DU) and linear trends (DU/decade and percent/decade) after applying MLR (see
section 2.2). Statistically significant trends are highlighted in bold (considered here as those within a 95% confidence interval).

| Vertical layer | Avg column (DU) | σ (DU) | Avg trend (DU/dec) | σ (DU/dec) | Avg trend (%/dec) | σ (%/dec) |
|---|---|---|---|---|---|---|
| Z < 2.5 | 7.5 | 1.9 | 0.12 | 0.28 | 1.6 | 3.7 |
| 2.5 < Z < 7.5 | 16.4 | 4.2 | **0.97** | **0.40** | **5.9** | **2.4** |



| 7.5 < Z < 12.5 | 20.8 | 12.6 | **0.52** | **0.19** | **2.5** | **0.9** |
|---|---|---|---|---|---|---|
| 12.5 < Z < 17.5 | 36.6 | 15.8 | 0.61 | 0.21 | 1.7 | 0.9 |
| 17.5 < Z < 22.5 | 64.9 | 13.6 | -1.38 | 0.81 | -2.2 | 1.3 |
| 22.5 < Z < 27.5 | 71.9 | 11.6 | -1.84 | 1.38 | -2.6 | 1.9 |
| 27.5 < Z < 32.5 | 48.9 | 12.2 | 0.20 | 0.39 | 0.4 | 0.8 |
| 32.5 < Z < 37.5 | 17.4 | 12.9 | **1.02** | **0.74** | **5.9** | **4.2** |

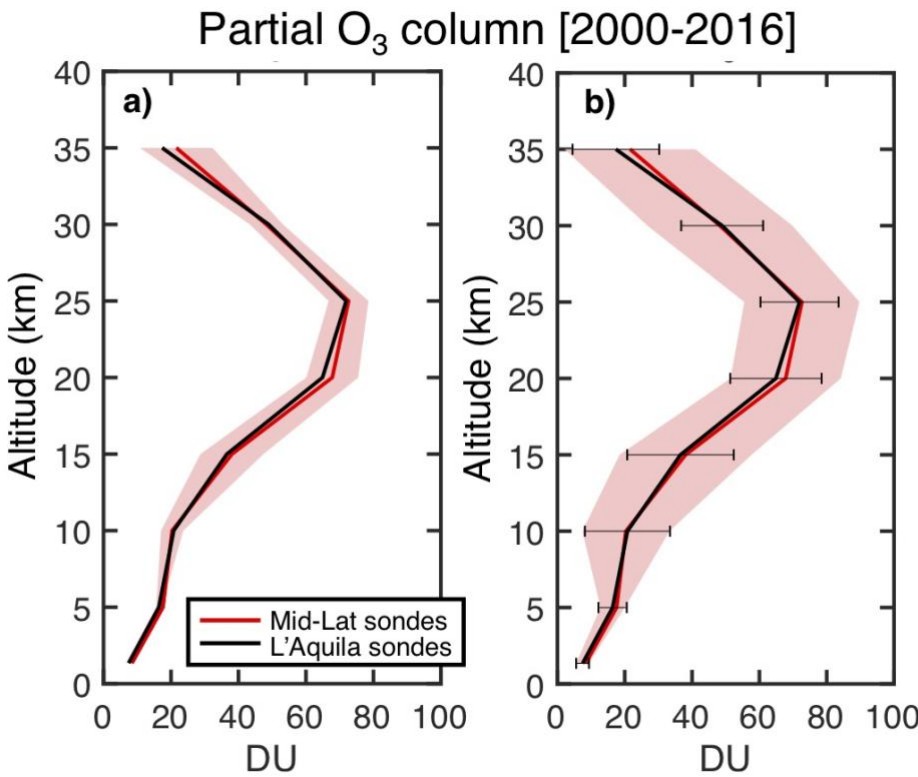

**Fig. 2.** Vertical profiles of partial O3 columns calculated for the layers in Tables 1-2 (average on years 2000-2016): black and red lines are for L'Aquila and other northern mid-latitude sounding stations, respectively (as in Fig. 1). The shaded area in panel (a) shows ±1σ of the variability of mean values among the 9 stations listed in Fig. 1, whereas in panel (b) it shows ±1σ of the partial column time variability in these same stations. The black whiskers are for ±1σ of the partial column time variability at L'Aquila.



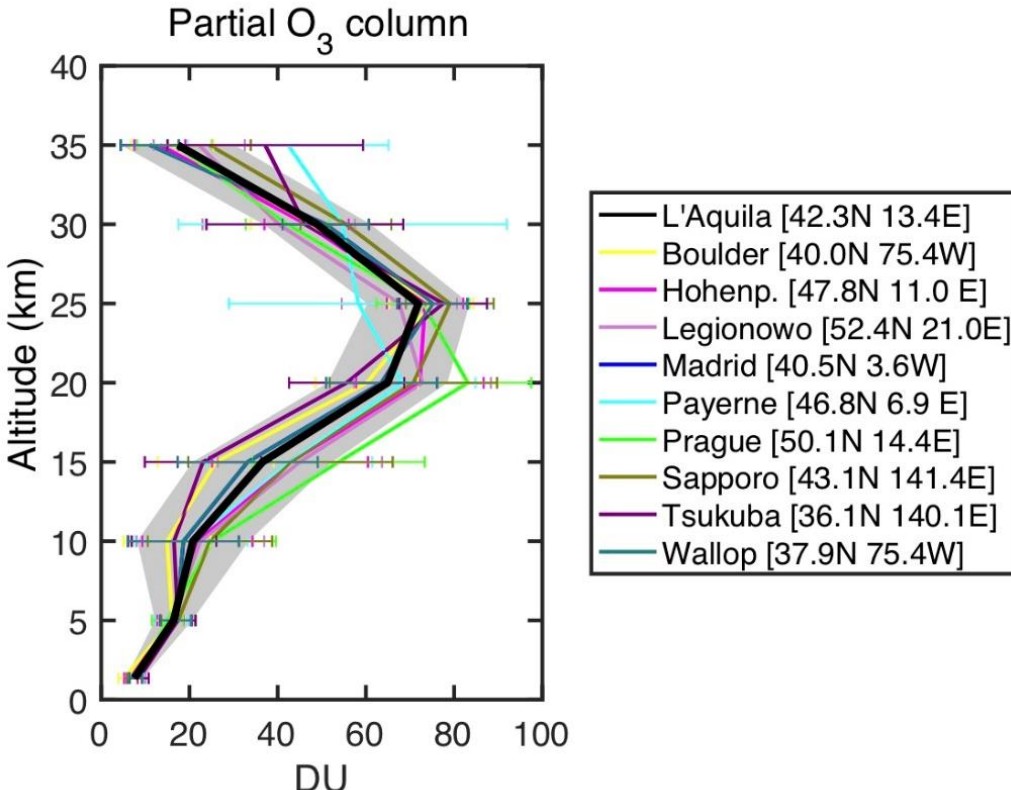

**Fig. 3.** As in panel (b) of Fig. 2 for the shaded area and the black line of the partial columns from L'Aquila soundings. The other lines refer to years 2000-2016 mean values at the other northern mid-latitude sounding stations (see legend) (Tilmes et al., 2012). Colored whiskers show the time variability at the single stations.


### 2.2 Ozone trend analyses

In order to maintain consistency with the results presented in the WMO (2014) Scientific Assessment of Ozone Depletion, a multiple linear regression (MLR) method has been applied to the ozonesonde measurements taken in L'Aquila, for deriving the trends shown in Table 2 and the following figures. This is done in order to remove natural variability due to the

following factors: QBO, ENSO, solar cycle and seasonal variability.

In Fig. 4 we show the time distribution of the soundings in L'Aquila for the partial columns described above. While data is sparser in the first 5 years of retrieval, we chose to include it anyway since results for the linear trend derived do not change significantly by excluding them. Superimposed to the single data points, we show the linear trend (after applying a multivalue linear regression described above) for all partial columns, separating between a positive slope (red lines) and a

negative one (blue line).





**Fig. 4.** Time series of partial O₃ columns from L'Aquila soundings (year and DU on x and y axis, respectively). Black circles
represent the data, while the superimposed red/blue lines (positive/negative slope, respectively) show a linear fit of the data
with the different factors removed by the MLR fitting (see text).

A summary of ozone trends in the northern mid-latitudes (35N-60N) starting in 1997 is reported in the WMO (2014) ozone
assessment (all numbers are given in percent/decade with ±2σ uncertainty): -4.6±2.2 at 70 hPa, -3.0±1.5 at 10 hPa, -6.8±1.8
at 2 hPa and -3.3±1.4 on the total column. These observed trends are based on multiple observational datasets (satellites,
ozone soundings, lidar, Umkehr, etc.) with a multi-linear regression accounting for QBO, solar cycle, volcanic aerosol and





ENSO. Previous ozone assessments had established the zonal mean patterns of ozone decline in the last three decades of the

20th century, due to increasing ODSs; the largest decline (6-8%) was found close to 2 hPa (42 km altitude) (WMO, 2011).

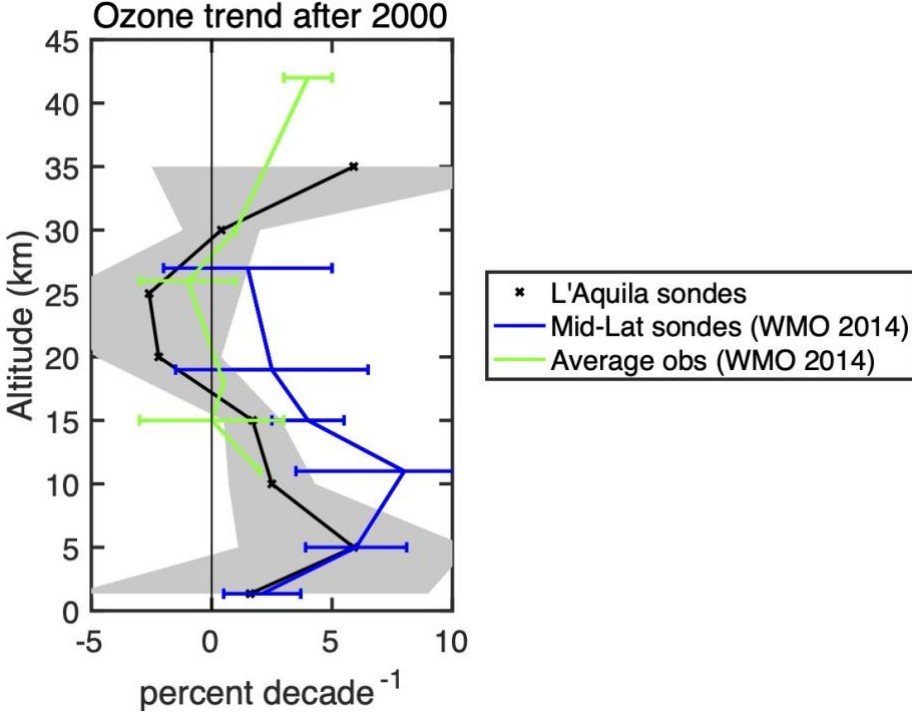


**Fig. 5.** Ozone trends in the layers of Tables 1-2 (percent/decade). The black line is for the linear trends obtained from L'Aquila partial column data; the shaded area shows ±2σ of the linear trend (see Table 2). The blue line is for other northern mid-latitude sounding stations (redrawn from WMO, 2014). The green line shows the ozone trend derived from multiple

observational datasets (i.e., satellites, ozone soundings, lidar, Umkehr, etc.) (redrawn from WMO, 2014; whiskers here highlight ±2σ uncertainty).

In Fig. 5 we compare the results derived in Fig. 4 with those described in the WMO report, both for mid-latitudinal soundings and for all observations. The results seem to agree in terms of positive/negative trend from the surface to the

troposphere, but to diverge in the lower stratosphere, while the mid-stratosphere decrease and the uppermost stratosphere increase seem to be consistent with the average of all observations. Large uncertainties are however present both in the reported Mid-Lat soundings and in the trend as calculated for the L'Aquila soundings.



## 3 Discussion of trends using CCMI models results

The statistically significant ozone trends highlighted in time series at L'Aquila are those in the mid-upper stratosphere (+5.5±4.2 percent/decade), mid troposphere (+5.9±2.4 percent/decade) and upper troposphere (+2.5±0.9 percent/decade). The UTLS positive trend (+1.7±0.9 percent/decade) is driven dynamically by the same mechanisms that regulate the upper tropospheric trend (as explained below in this section), but from our data analysis it does not appear to be statistically significant, contrary to those in the mid and upper troposphere.

The upper stratospheric positive trend was already well documented in the WMO (2014) ozone assessment (+3.9±1.3 percent/decade, here with ±2σ uncertainty). It may be attributed to the following concomitant causes: (a) decrease of total organic chlorine and bromine (Fig. 6ab), with associated reductions of $Cl_y$ and $Br_y$ (Fig. 6cd); (b) increases of $CO_2$, $CH_4$, $N_2O$ (Fig. 7abc) and $H_2O$ (Fig. 8a), with an associated decrease of stratospheric temperatures due to more cooling to space (Fig. 7d and Fig. 8); the increasing amount of stratospheric $H_2O$ results from the TTL warming and increasing $CH_4$

oxidation. These radiative-chemical mechanisms affecting upper stratospheric ozone are pretty well known and were widely discussed in past WMO assessments (WMO, 2011; WMO, 2014; WMO, 2018). They can be summarized as a slowing down of the catalytic chemical cycles for ozone destruction involving gas phase homogeneous reactions, because of the negative temperature trend (*i.e.,* more cooling to space produced by the increasing $CO_2$ and other GHGs). $Cl_y$ and $Br_y$ catalytic cycle in the upper stratosphere, in turn, are slowed down due to the decreasing amount of organic chlorine and bromine.

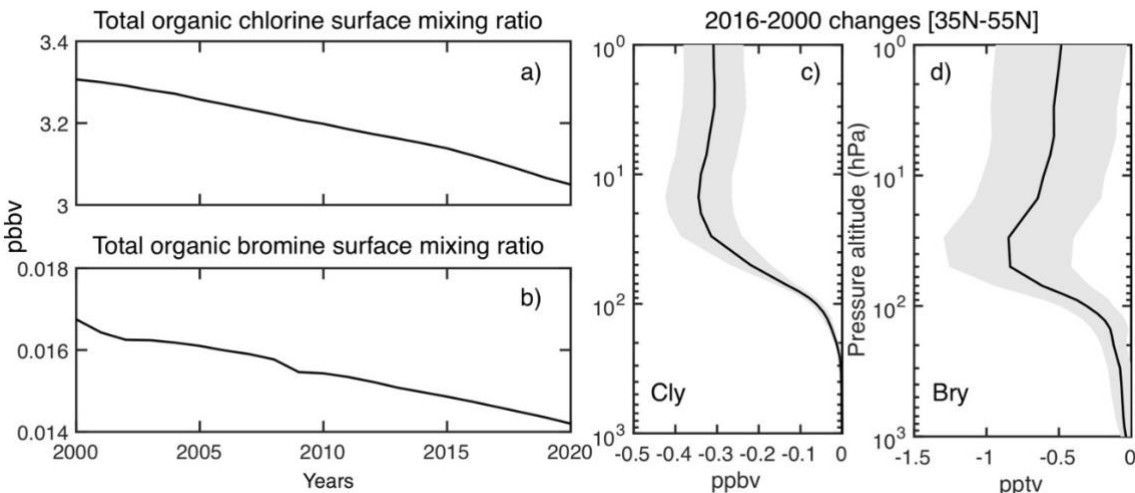


**Fig. 6.** Time series (2000-2020) of surface mixing ratios of total organic chlorine (a) and bromine (b); values are those used as prescribed boundary conditions for the CCMI RefC2 numerical experiments (Eyring et al., 2013). Panels (c,d): vertical profiles of northern mid-latitude changes (35N-55N) of total inorganic chlorine ($Cl_y$, ppbv) (c) and total inorganic bromine ($Br_y$, pptv) (d) (2016-2000) averaged over all available models for refC2 (black line). Part of the models explicitly include

the contribution of $CH_2Br_2$ and $CHBr_3$ to the Bry budget, other use a 5 pptv increase of the $CH_3Br$ surface mixing ratio as a





proxy for these VSLS (see Morgenstern et al., 2017). The corresponding shading represents ±1 σ of the ensemble of models. The profiles for the single models are reported in Fig. S1.

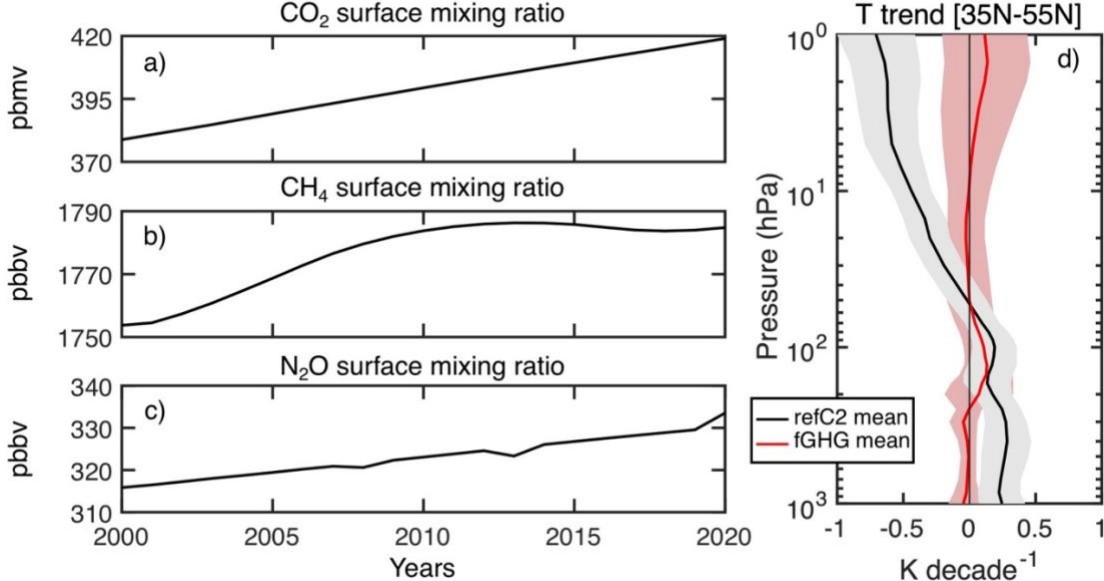

**Fig. 7.** Time series (2000-2020) of surface mixing ratios of $CO_2$ (a), $CH_4$ (b) and $N_2O$ (c); values are those used as prescribed
boundary conditions for the CCMI RefC2 numerical experiments (Eyring et al., 2013). Panel (d): vertical profiles of northern mid-latitude trends (35N-55N) of temperature (K/decade) averaged over all available models for refC2 (black line) and fGHG (red line). The corresponding shadings represent ±1 σ of the ensemble of models. The profiles for the single models are reported in Fig. S2.

For changes in the troposphere, the multi-model study of Stevenson et al. (2006) pointed out the importance of climate
change for the tropospheric ozone burden. In that study, the ozone chemical precursors were kept fixed in a perturbed experiment operated with year 2030 climate conditions and in a baseline reference case with year 2000 climate. They found a negative climate feedback dominating at low altitudes, which they attributed to higher humidity and hence increased ozone destruction via the reaction $O(_1D) + H_2O$. At the same time, they found upper tropospheric ozone to increase, especially in the Northern Hemisphere, and this was attributed to an increased influx from the stratosphere. They concluded that these two
climate feedbacks (water vapor and injection from the stratosphere) appear to be the dominant mechanisms operating.
These mechanisms apply to our results as well, together with the changes in ozone chemical precursors during the seventeen years. The time evolution of mean zonal tropospheric $NO_x$ and $HO_2$ at northern mid-latitudes from CCMI simulations does not suggest a potential significant impact on in-situ production of tropospheric $O_3$, with the exception of the increasing amount of $CH_3O_2$ from $CH_4$ (Fig. 7b). The first of the two climate feedbacks discussed before can be easily observed in the
tropospheric $H_2O$ increase of Fig. 7d; however, to precisely determine the contribution of the fluxes from the stratosphere at





mid-latitudes, three distinct elements are necessary: a) the increased mid-latitudinal stratospheric ozone mixing ratio due to increased horizontal isentropic transport from the main photochemical source region; b) the increased mid-latitudinal stratospheric downwelling fluxes and finally c) the increase in Stratosphere-Troposphere Exchange (STE) fluxes. We show these three contributions to the increase of upper-tropospheric ozone in Fig. 9 (for stratospheric horizontal and vertical mass

fluxes) and Fig. 10 (for STE fluxes). Both horizontal and vertical fluxes are calculated starting from the velocities, ozone concentration and temperature 3D monthly-averaged (as done already by Visioni et al., 2017) fields made available by each CCMI group, interpolated on a common vertical grid (considering that different models have different vertical coordinates, as explained by Morgenstern et al., 2017). The use of the monthly averaged fields only allows an evaluation of the large scale and not of the eddy-driven fluxes: however, considering our goal of evaluating the trends in these quantities between

the two time periods, whatever error might be introduced by this calculation is not influent to our overall conclusions.

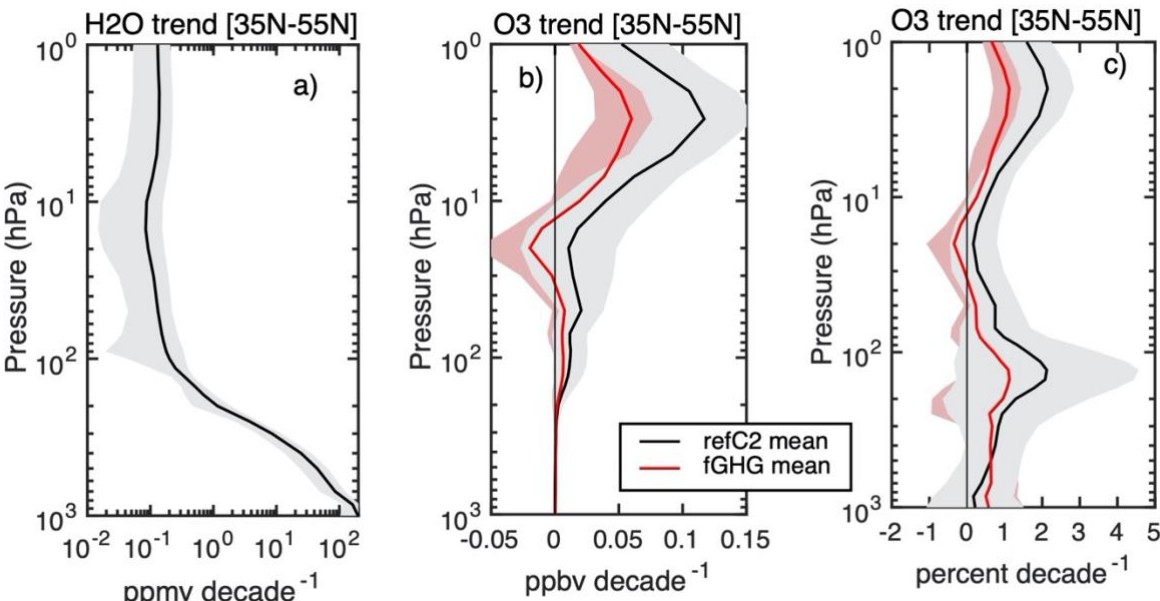

**Fig. 8.** Vertical profiles of northern mid-latitude trends (35N-55N) of $H_2O$ (ppmv/decade) (a) and ozone (ppbv/decade in (b) and percent/decade in (c)) averaged over all available models for refC2 (black line) and fGHG (red line). The corresponding shadings represent ±1 σ of the ensemble of models. The profiles for the single models are reported in Fig. S3 (for panel a)

and Fig. S4 (for panels b-c).

In Fig. 9 we compare both the vertical and horizontal $O_3$ fluxes averaged over all available CCMI models between 10 and 100 hPa. In this figure we also show results for the CCMI simulations with fixed Ozone Depleting Substances (fODS): this is done to check if, for the period we are considering, changes driven by ODS are significant. Fig. 9a and Fig. 9b show the base fluxes averaged over the years 1999 and 2000 for refC2, fGHG and fODS, showing an overall agreement of all models





regarding the mid-latitudinal stratospheric sign of the fluxes: a downwelling for the vertical flux and a poleward movement for the horizontal fluxes. The changes in the 2000 to 2016 period are shown in Fig. 9c and Fig 9d: whilst different models predict different changes in value, all models show consistently a strengthening of the BDC due to the increase of GHGs in the refC2 simulation that result, for the analyzed region, in an increased (i.e., more negative) downwelling and an increased poleward transport. This is further confirmed by the absence, in the fGHG simulations, of significant changes in the fluxes.

Lastly, in Fig. 9e and Fig. 9f, we show these changes in terms of the base fluxes, that result in an average 8.2±5.1% increase in downwelling and an average 4.1±2.5% increase in poleward transport. The similarities in the response between the refC2 and fODS show that, unlike the analyses presented by Polvani et al. (2018) (where the authors looked at changes at the end of the 21st century), for the period under our consideration changes due to ozone depleting substances are negligible compared to those due to greenhouse gases.

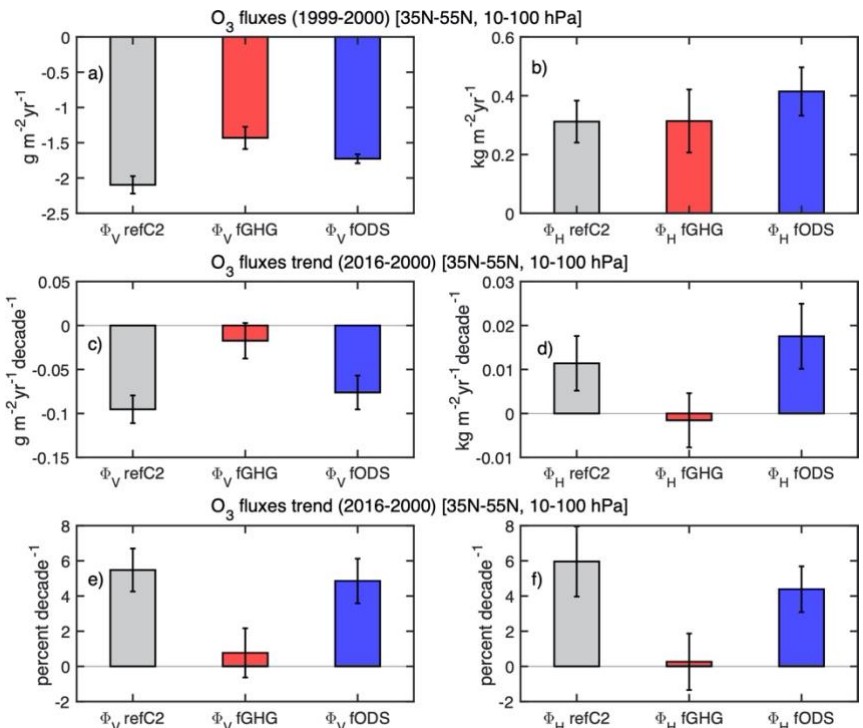

**Fig. 9.** a) Left panels: a) mid-latitude LS vertical (g/m2/yr) fluxes of O₃ (1999-2000) for refC2 (grey bar, average over 9 models) and fGHG (red bar, average over 6 models). The error bars represent ±1 σ of the ensemble of values for all models. c) Linear trend (g/m2/yr/decade) of the LS vertical fluxes of O₃ between 2000 and 2016 for the same models. e) Same as c) but expressed in % of the fluxes in panel a). The values for all separated models are shown in Fig. S5. Right panels: b) mid-latitude LS horizontal (kg/m2/yr) fluxes of O₃ (1999-2000) for refC2 (grey bar, average over 15 models) and fGHG (red bar, average over 8 models). d) Linear trend (kg/m2/yr/decade) of the LS horizontal fluxes of O₃ between 2000 and 2016 for the





same models. e) Same as d) but expressed in % of the fluxes in panel b). The values for all separated models are shown in Fig. S6.

Finally, the enhanced stratospheric influx is well captured in Fig. 10, in the analyses of the trends of the ozone STE from the ULAQ-CCM. This ozone flux is calculated following the specific recommendations available at the SPARC-CCMI website for simple tracer experiments (https://blogs.reading.ac.uk/ccmi/data-requests-and-formats/). The guidelines for the 'o3ste' tracer experiments are the following: ozone tracer intended to map out the stratosphere-troposphere exchange of ozone: (a) o3ste is set to $O_3$ at the model top layer; (b) standard $O_3$ production-loss terms are used to change o3ste every chemistry time

step for 'stratospheric' air grid boxes; (c) o3ste is reset to $O_3$ in the lowermost layers (fix these layers to be up to sigma=0.9, ~1 km altitude): this tropospheric reset is recorded as a 2D (latitude-longitude) monthly mean tendency (Tg per month per grid cell); (d) 'stratospheric' air is determined in any way that is most convenient (e90, $O_3$, PV-theta, etc). The ULAQ-CCM was the only one CCMI contributing model running this specific tracer experiment.

At mid-latitudes, the STE has been connected with deep convection (Hegglin et al., 2004; Tang et al., 2011) and is an

important term in the budget of tropospheric ozone (Voulgarakis et al., 2010). An increase in downward $O_3$ fluxes is observed in the northern mid-latitudes (Fig. 10a) especially over the continents where this flux is, on average, maximal (Fig. S7). This positive trend is shown to be especially significant over the months of April, May and June (where the flux is also stronger), as shown in Fig. 10b. These results confirm previous projection of STE changes due to the increase in GHG concentrations (Sudo et al., 2003; Eyring et al., 2007; Oman et al., 2010). The decrease of the same STE fluxes for the fGHG

simulation shown in Fig. 10c and Fig. 10d is on the other hand simply due to the decrease in ozone-depleting substances, that is not balanced and compensated by the transport changes discussed for the refC2 case.





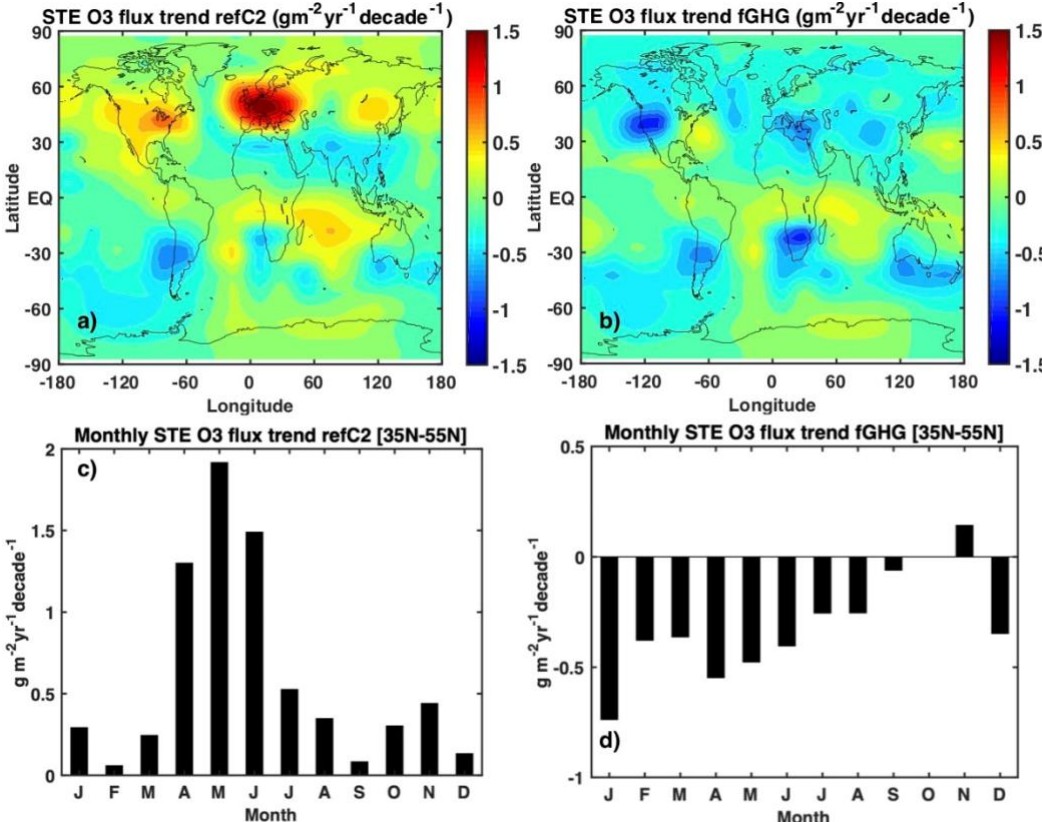

**Fig. 10.** a) STE O₃ flux linear trend (g/m2/yr/decade) between the years 2000 and 2016 for the ULAQ-CCM refC2

simulation. b) Same as in a), but for the fGHG simulation. c) Monthly STE O₃ flux linear trend (g/m2/yr/decade) averaged

between 35N and 55N for the refC2 simulation. d) Same as in c), but for the fGHG simulation.

In Fig. 11 we show a comparison of the trend observed by the soundings in L'Aquila and the overall trend predicted by the

CCMI models in the same period, as in Fig. 8c, calculated in the same partial column described in Table 2. While the

soundings usually overestimate both the tropospheric increase and the upper stratospheric increase compared to the models,

a general agreement is present in both with regards to the general sign of the trend: positive in the uppermost stratosphere

due to the cooling produced by the increase in greenhouse gases and positive near the surface.

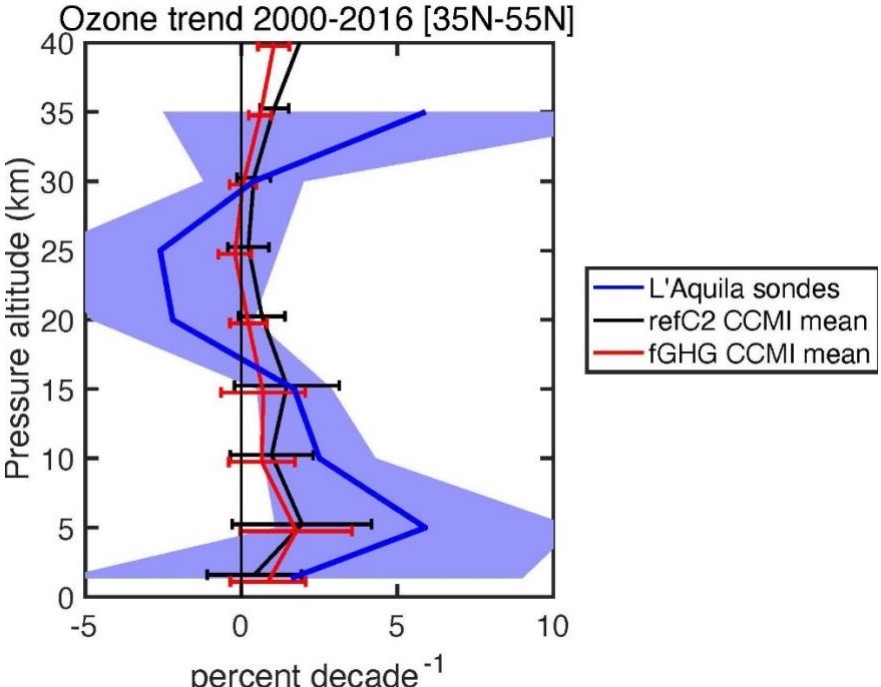

**Fig. 11.** Ozone trends in the layers of Tables 1-2 (percent/decade), with blue line and shading as in Fig. 5 (L'Aquila sounding). Black and red lines show the northern mid-latitude trends (35N-55N) derived from CCMI model runs, for cases RefC2 and fGHG, respectively.

## 4 Conclusions

We have reported in this work the results from over 200 ozone soundings-derived vertical profiles measured in the town of L'Aquila (42.3°N, 13.4° E) between 2000 and 2016, and never published before. The results available from other locations in the northern mid-latitudes have been compared with our results showing a good agreement both in terms of the overall time averaged $O_3$ partial column and with the trends already reported in the WMO report (WMO, 2014). These trends show a similar increase in $O_3$ in the layers ranging from the surface to the uppermost stratosphere, while a decrease is reported in our trends between 16 and 30 km that, while in disagreement with the other reported ozone soundings, is more similar to the overall trends reported from a wider array of instruments.

To further understand the observed trends in the measurements we used the modeling results from the Chemistry-Climate Models Intercomparison (CCMI) (Morgenstern et al., 2017). In particular, we focused on the reference simulation refC2 (with greenhouse gases that follow the RCP6.0 IPCC scenario (Meinshausen et al., 2011) to understand the 2000-2016 trend



and the scenario with greenhouse gases concentration fixed at 1960 levels (fGHG) to understand the role of the increased surface warming (and connected stratospheric cooling) that can be revealed by looking at the difference between refC2 and fGHG.

Changes in mid-latitudinal high tropospheric ozone are mainly due to two different factors: changes in the concentration of ozone-depleting substances and modifications of the stratospheric transport and of the stratospheric-tropospheric connections

that result in different influxes of $O_3$ from the stratosphere. For the former, we show that, since the global surface mixing ratio in the CCMI models of chlorine and bromine substances is prescribed to decrease in the considered time period (Eyring et al., 2013; Chipperfield et al., 2013), the vertical profiles also show a substantial decrease in their concentration: this would, in turn, result in less ozone depletion at mid-latitudes. Together with this, however, we show in the CCMI models an overall increase in mid-latitudinal humidity due to temperature changes in the stratosphere that would enhance ozone

destruction via the reaction $O(_1D) + H_2O$.

For the latter, we prove that all models tend to agree over the sign of the changes in mid-latitudinal transport for the considered period: all models detect (albeit with differences in the magnitude of the predicted changes) a strengthening in the mid-latitudinal downwelling, coupled to an increase in poleward transport from the tropical area, where most of the ozone is produced. For one of the models (ULAQ-CCM) we show that these two factors are then coupled to an increase in

Stratospheric-Tropospheric Exchange fluxes, further confirming the positive trend of tropospheric ozone that is detected by most instruments, and in particular by the L'Aquila ozonosondes, over the same time period at those latitudes.

**Data availability.** Observations data are available from the ozone sounding team at the University of L'Aquila (vincenzo.rizi@aquila.infn.it); data from other soundings is available at the World Ozone and Ultraviolet Data Center

(http://www.woudc.org/). All data from CCMI-1 used in this study can be obtained through the British Atmospheric Data Centre (BADC) archive (ftp://ftp.ceda.ac.uk, last access: 15 March 2018). CESM1-WACCM and CESM1-CAM4 data were downloaded from http: //www.earthsystemgrid.org. For instructions for access to both archives see http://blogs.reading.ac.uk/ccmi/badc-data-access or Hegglin et al. (2015).

**Acknowledgments.** The EMAC simulations contributed to CCMI-1 have been performed at the German Climate Computing Centre (DKRZ) through support from the Bundesministerium für Bildung und Forschun (BMBF). DKRZ and its scientific steering committee are gratefully acknowledged for providing the HPC and data archiving resources for this consortial project ESCiMo (Earth System Chemistry integrated Modelling). The CCSRNIES simulations were performed on NEC-SX9/A(ECO) and NEC-SXACE computers at the Center for Global Environmental Research (CGER), National Institute for

Environmental Studies (NIES), and supported by the Environment Research and Technology Development Fund, Ministry of Environment, Japan (2-1303 and 2-1709) and Japan Society for the Promotion of Science (JSPS) KAKENHI Grant Number 20H01977.



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
