# Peer review of "Seventeen years of ozone sounding at L'Aquila, Italy: evidence of mid-latitude stratospheric ozone recovery and tropospheric profile changes"

_Atmospheric Chemistry and Physics, 2020_

## Referee Comment (RC1) · Anonymous Referee #1 · 22 Jul 2020

GENERAL COMMENTS

This paper presents an analysis of trends in vertical ozone profiles obtained from 295 ozonesonde flights made at L-Aquila, Italy from 2000 to 2016. A partial attribution of the drivers of observed changes is then made through a diagnosis of transient simulations of several chemistry-climate models (CCMs) made available through the SPARC CCMI project. Simulations with GHG concentrations fixed at 1960 levels (fGHG) and with ozone depleting substance concentrations fixed at 1960 levels (fODS) are used together with the reference simulations (refC2) to understand how different chemical and

dynamical drivers of ozone changes at Northern mid-latitudes have affected ozone over L'Aquila this century.

In its current state the paper is not suitable for publication in ACP for the reasons outlined below. I would, however, strongly encourage the authors to revise the paper taking into consideration my suggested changes because I believe that this analysis does have merit and value. It may only be that some key information is missing in a few places.

SPECIFIC COMMENTS

Line 61: Tropospheric influx of what? Similarly on line 64 you mention 'enhanced stratospheric input'. Input of what? Water vapour, ozone?

Line 72: But ozone recovery is a very different thing to ozone increases. Let me give you an example: even though ozone in the tropical lower stratosphere is decreasing, it is recovering from the effects of ODSs. I have had to make this point in reviewing several papers. Here is, verbatim, what I said in response to an earlier different paper I reviewed:

"Could I put it to you that the Montreal Protocol has been effective in reducing ODS concentrations, and thereby concentrations of Cly and Bry throughout the atmosphere, and that, as a result, ozone throughout the atmosphere, including the lower stratosphere, is recovering from the effects of those ODSs. Is this recovery apparent in observations in the upper stratosphere? Apparently yes. I say apparently only in that (at least in this paper) a thorough attribution of the drivers of those ozone increases has not been done. Is this recovery apparent in observations in the lower stratosphere? No, clearly not. Why not? Well because other factors have been affecting ozone (not diagnosed in this paper) that are likely (we cannot be sure since a thorough attribution has not been done) overwhelming the increases brought about by reductions in concentrations of Cly and Bry. Wouldn't that be a more accurate picture to communicate to policy-makers?"

The point I need to make here is that ozone increases are not indicative of recovery (from the effects of ODSs). Ozone increases aren't even a prerequisite for ozone to be recovering. What is an imperative for detecting ozone recovery is an *attribution* of ozone increases to decreases in concentrations of ODSs, even if those ozone increases are offset by other factors. That attribution can be done, in large part, by comparing the fODS and refC2 CCMI simulations as you have done. So all I am saying is be careful of the word 'recovery'. Use it when you really do mean recovery from the effects of ODSs, but don't use it as a synonym for ozone increases.

Line 125: This is the entire *mean* profile, averaged over all 295 flights right?

Figure 1: Rather than having two panels couldn't you just plot the ozone mixing ratios on a log scale?

Table 1: Why not just use the GPS altitude from the radiosondes instead of calculating a pressure altitude? If you are going to calculate a pressure altitude, shouldn't you at least use H=RT/g so that you have a temperature dependent scale height.

Line 165: In calculating the uncertainties on the trends, did you account for the effects of autocorrelation in the regression residuals? If not, why not?

Figure 4: Has the seasonal cycle been removed from these data? If so, then please say so and describe how. If not, then why do I not see a seasonal cycle in the partial columns? It is not clear from the figure caption whether the MLR has been used to remove the variability from the measurements or from the regression mode fit. This needs to be explained much more clearly. I would like to see the formula for the MLR, a description of how the annual cycle was removed (fitting a Fourier expansion?), how seasonality in the fit coefficients was handled (e.g. the QBO affects ozone more in some seasons than in others) and how autocorrelation in the residuals was accounted for in the calculation of the trend uncertainties. No explanation is provided as to how the uncertainties were calculated.

Lines 178-181: There are two things that confuse me here: 1) Over what period are these trends calculated? You state that the trends are from 1997 but you give no end year. 2) I am very surprised that the trends are all negative. From everything I have read, since 1997 ozone has been increasing in the northern mid-latitude middle and upper stratosphere. The values that you quote also seem to not correspond at all to the values plotted in Figure 5 (blue line).

Line 194: But the values for 'all observations' are also just for 35N-60N right?

Line 194: Why only 'seem to agree'? Surely you can calculate the degree of statistical agreement and provide a quantitative value that describes the level of agreement and don't have to resort to vague statements such as 'seem to agree'?

Line 209: You need to define the TTL acronym.

Figure 6: Previously you were taking 35N-60N to be the northern mid-latitudes but now you are taking 35N-55N. Why the change?

Line 235: Injection of what from the stratosphere? Ozone I presume, but you need to avoid the reader having to presume.

Line 243: Again, I assume you are referring to ozone fluxes here?

Figure 8: Somewhere, either in the figure or in the figure caption it needs to be stated over what period these trends were calculated.

Figure 9: No reference is made in the figure caption as to what is shown in panel (f) - I think that you mistakenly refer to panel (e) rather than (f).

Line 287: You need to expand on what the 'e90, O3, PV-theta' refer to. Do you really want readers to have to guess what e90 is?

GRAMMAR AND TYPOGRAPHICAL ERRORS

I understand that the first author's first language may not be English and, as a result,

there will possibly be some grammatical errors in the paper. This should not detract from the scientific quality of the research. I have made some suggestions in places for where these might be corrected but feel it is not my role to copy edit the paper. What does annoy me somewhat is that there are many co-authors on this paper whose first language is English and it is clear that they didn't even bother to proof read this paper and suggest corrections to the grammatical errors. My advice to the lead author would be to get some of the co-authors to at least do that much.

Line 40: Replace 'associated to higher' with 'associated with higher'.

Line 42: Replace 'do not result to be statistically significant in our analyses' with 'are not statistically significant in our analyses'.

Line 48: Replace 'emerging evidences' with 'emerging evidence'.

Line 50: Replace 'as SPARC-CCMVal' with 'such as SPARC-CCMVal'.

Line 56: Replace 'greenhouse gases (GHGs)' with 'greenhouse gas (GHG) concentrations'.

Line 60: Replace 'increase out of the tropics' with 'increase poleward of the tropics'.

Line 63: Replace 'for year 2030' with 'for 2030'.

Line 68: Replace 'year 2000 to the year 2016' with '2000 to 2016'.

Line 85: Replace 'coupled' with 'ozonesonde coupled'.

Line 89: Replace 'preparation's procedure' with 'preparation procedures'.

Line 117: Shouldn't this be 'Wallops Island' rather than 'Wallop'? Likewise on line 122.

Line 122: Replace 'ozonosondes data' with 'ozonesonde data'.

Line 135: Replace 'errorbar' with 'error bar' throughout.

Line 136: Replace 'single sondes location' with 'single sonde location'.

Line 213: Replace 'catalytic cycle' with 'catalytic cycles'.

Line 250: Replace 'is not influent to our overall' with 'does not influence our overall'.

---

## Referee Comment (RC2) · Anonymous Referee #2 · 30 Jul 2020

Review of acp-2020-525

Seventeen years of ozone soundings at L'Aquila, Italy: evidence of mid-latitude stratospheric ozone recovery and tropospheric profile changes by Daniele Visioni et al.

The paper by Visioni et al. presents 17 years of ozone profile measurements at the station of L'Aquila (Italy), which are used for trend analyses of partial columns ranging from the boundary layer up to the middle stratosphere. The results are discussed in the light of enhanced greenhouse gas concentrations and the corresponding changes

of dynamical and chemical processes in recent years, for instance the acceleration of the Brewer-Dobson circulation and the slowing down of ozone destroying gas-phase reactions due to stratospheric cooling. Results of Chemistry-Climate Models (CCMs) are consulted to point out the role of the most important dynamical and chemical processes. The manuscript in its present form cannot be published in ACP. Currently it contains several (interesting) points, but which are not matching well with each other, in other words, the "red thread" cannot be seen in this paper. I am missing concrete scientific results and further pointing out the advantages of this new, long-term ozone time series of L'Aquila (for instance with respect to local tropospheric ozone changes, i.e. differences between the other mid-latitude ozone stations in the Northern hemisphere). The presented draft needs major revisions, not only with respect to the structure of the scientific paper, but also regarding the scientific content including the interpretation and the discussion of results. Moreover, the used methods of analyses are not adequately explained or they are not referring to the respective literature. So far the scientific significance of this study is low, the scientific quality is fair and the presentation quality of the paper is good.

The scientific content of this paper should focus on what have been promised in the title: first, provide evidence for mid-latitude ozone recovery and second, explanation of changes of tropospheric ozone profiles. In a first step (section) you should introduce in more detail the presented new ozone data set of L'Aquila (which "have never been published before in the scientific literature", page 3, line 68), by describing (briefly) the used measurement system. How do you organize the quality check of the ozone data, and how does the data set compare with other (Northern mid-latitude) ozone stations (for instance more detailed discussions of deviations and the range of uncertainty are needed). In a next step (section) you should describe the methods of data analyses, which are used in the paper. Afterwards you can present the results of your trend analysis, which should be compared in more detail to the findings at other Northern mid-latitude stations (based on your discussion about Figs 2 and 3; and possibly a comparison with satellite measurements would be nice). This will help to classify much

better the measurements at L'Aquila in comparison with other stations, i.e. showing that the data of L'Aquila are (more or less) in line with other respective ozone data sets. Finally, for the interpretation of the trend analyses, results of specific CCM simulations can help to explain the role important atmospheric processes, which are responsible for the observed changes. At the end the major (new) findings of this study should be summarized (or at least say that your results are in agreement with recent (other) investigations). Currently in the available manuscript I cannot see results or conclusions, which have not been published before.

Here are, in more detail, my major points:

- It would be helpful to present first the (new) ozone data set of L'Aquila including a more detailed description of the measurements (including the technical description of the used instruments) and how you have verified and validated your measurements. And some words about the long-term stability of the 17-year time series of ozone data would be helpful. In another section you should (briefly) describe your method of trend analyses and explain your statistical analysis (which is the foundation of your investigations). At least the corresponding literature should be cited. And then, in a next section, you can present the results of your trend analyses and discuss how robust the ozone trends at Northern mid-latitudes are. Here it would be highly desirable to have a comparison with the trends of the different partial columns at the different Northern mid-latitude ozone sonde stations. It would be nice to see if the L'Aquila station shows similar or different trends, or which other stations show most obvious "differences". At the end of the trend section it would be nice to get an overview of the mid-latitude ozone trends and a statement about the robustness of the "sign of ozone recovery" in the Northern hemisphere. A more detailed trend analysis (I am hoping for an "improved" statements about ozone recovery at Northern mid-latitudes) would guarantee a significant upgrade of this study. In case of a consistent picture, in a next step results of specific CCM simulations may be used to explain the identified trends. I like very much the idea using the results of CCM simulations, but I think that you have used the

wrong set of CCM-simulations (see my next comment below) for the analysis of the recent ozone trends (2000 to 2016).

- For the discussion of CCM results you are using the refC2 simulation, which is the reference forecast simulation (the phrase "historical simulations (refC2)", page 3, line 73, is misleading). Although refC2 is running from 1960 (until 2100), it does contain boundary conditions for the past, which are not based on observations (e.g. observed sea-surface temperature (SST) and sea-ice cover (SIC), concentrations of greenhouse gases and other chemical compounds and emissions). To make a consistent future simulation refC2 (without a break when switching from the past into the future), for instance the SST field is taken from an ocean model or the greenhouse gas concentrations are following the RCP6.0 scenario. I think that for the comparison with the L'Aquila ozone data (and other Northern mid-latitude ozone stations) it would be better to use the available CCM results of the refC1SD (specified dynamics) or the refC1 (free running climate) simulations, which are both focusing on the recent past (e.g. from 1980 to 2017). Concentrating your investigations on the Northern mid-latitudes, a comparison with respective observation (mean conditions) with results of CCMs (multi-model means of refC1SD and refC1) would be much better. And in a last step you can in particular analyze the results of L'Aquila together with the results of other single ozone stations, and checking the importance of relevant atmospheric processes, which are influencing the ozone trends (profiles) in Northern mid-latitudes. My point is that you should look in more detail into the results (e.g., ozone fluxes from the stratosphere to the troposphere) of your analyses. So far the discussion regarding the CCMI model results is in larger parts superficial. Finally, in addition a look into the future (based on refC2, up to 2100) would make sense, also in combination with the sensitivity simulations (e.g., senC2-fGHG or -fODS).

- The role of the strengthening of the Brewer-Dobson circulation (BDC) (due to enhanced greenhouse gases) is mentioned and discussed. So far, this is a result of simulations (with enhanced greenhouse gas concentrations) with global circulation models

(including CCMs), but so far studies based on measurements in recent years (e.g., Engel et al., Nature, 2008; Fu et al., Environmental Research Letters, 2019) did not find an increase of the BDC (or a decrease of the mean age of stratospheric air), in particular after year 2000. Since you are looking in the time period from 2000 to 2016 there is an obvious discrepancy between assessments based on observations and model simulations (e.g., Butchart, Review of Geophysics, 2014). This point must be mentioned not only in the Abstract (page 2, lines 38-39) and Introduction (page 2, lines 57-58), but also in detail in the discussion-section (with respect to the ozone fluxes, especially the downward transport from the stratosphere into the troposphere) and also in the Conclusions (page 18, paragraph beginning in line 336). In particular, your findings regarding the refC2 ULAQ-CCM simulation (page 15, line 280), indicating enhanced stratospheric influx, need further discussions with respect to the observations. In general, so far the discussions of results regarding the relevant atmospheric processes are mostly qualitative. If the analyses of the CCM simulations would be intensified in this study, concrete results with more quantitative statements can be expected.

Some minor points:

- Page 3, line 67 and line 81: The lat/lon coordinates of the L'Aquila station should be the same. Please check it regarding consistent information in the manuscript.

- Page 3, line 68 and page 4, line 102/103: ". . . have never been published before. . ." is in contradiction to the statement ". . . L'Aquila ozone-sounding data were used in the validation . . . retrieved by MIPAS. . .". This should be clarified.

- Page 6, lines 129 to 138: This paragraph is hard to read; it is difficult to get the message of it. Please try to give a clearer statement.

- Page 11, line 200 and line 205: is the altitude range (mid-upper stratosphere and upper stratosphere) here the same? Are the estimated trend values comparable?

- As an example, page 11, line 210: if you are referring to the ozone assessments,

it would be helpful to cite also the respecting Chapter, where the information can be found.

- Page 12 and 13, lines 240 to 245: wrong Figure numbers in the text; Fig 7d should be Fig 8a (?!), Fig 9 should be Fig 8 and Fig 10 should be Fig 9. In the following the numbering of the figures is okay again.
* * *